# Mutual Coupling Reduction in Antenna Arrays Using Artificial Intelligence Approach and Inverse Neural Network Surrogates

**DOI:** 10.3390/s23167089

**Published:** 2023-08-10

**Authors:** Saeed Roshani, Slawomir Koziel, Salah I. Yahya, Muhammad Akmal Chaudhary, Yazeed Yasin Ghadi, Sobhan Roshani, Lukasz Golunski

**Affiliations:** 1Department of Electrical Engineering, Kermanshah Branch, Islamic Azad University, Kermanshah 67771, Iran; 2Department of Engineering, Reykjavik University, 102 Reykjavik, Iceland; 3Faculty of Electronics, Telecommunications and Informatics, Gdansk University of Technology, 80-233 Gdansk, Poland; 4Department of Communication and Computer Engineering, Cihan University-Erbil, Erbil 44001, Iraq; 5Department of Software Engineering, Faculty of Engineering, Koya University, Koya 46017, Iraq; 6College of Engineering and Information Technology, Ajman University, Ajman 346, United Arab Emirates; 7Software Engineering and Computer Science Department, Al Ain University, Al Ain 64141, United Arab Emirates

**Keywords:** antenna design, artificial intelligence, isolation, resonator, mutual coupling, numerical optimization, particle swarm optimization (PSO)

## Abstract

This paper presents a novel approach to reducing undesirable coupling in antenna arrays using custom-designed resonators and inverse surrogate modeling. To illustrate the concept, two standard patch antenna cells with 0.07λ edge-to-edge distance were designed and fabricated to operate at 2.45 GHz. A stepped-impedance resonator was applied between the antennas to suppress their mutual coupling. For the first time, the optimum values of the resonator geometry parameters were obtained using the proposed inverse artificial neural network (ANN) model, constructed from the sampled EM-simulation data of the system, and trained using the particle swarm optimization (PSO) algorithm. The inverse ANN surrogate directly yields the optimum resonator dimensions based on the target values of its S-parameters being the input parameters of the model. The involvement of surrogate modeling also contributes to the acceleration of the design process, as the array does not need to undergo direct EM-driven optimization. The obtained results indicate a remarkable cancellation of the surface currents between two antennas at their operating frequency, which translates into isolation as high as −46.2 dB at 2.45 GHz, corresponding to over 37 dB improvement as compared to the conventional setup.

## 1. Introduction

Microstrip patch antennas have been widely used in the recent communication systems due to their low cost and easy fabrication [1]. Utilization of patch antennas has grown rapidly with the development of new application areas, such as wearable devices [2,3], 5G communications [4,5,6,7,8], internet of things (IoT) [9,10], optical systems [11,12,13], biomedical engineering [14,15,16,17] and multiple input multiple output (MIMO) systems [7,18,19,20].

High isolation between the elements of patch antenna arrays is an important consideration, especially in the MIMO systems [21,22]. High mutual coupling (MC) between the patch antennas decreases the efficiency and performance, particularly in terms of diversity gain [23,24]. Over the past decade, several methods and decoupling structures have been introduced to overcome this drawback. The recently proposed methods for MC reduction in antenna arrays include, among others, utilization of defected ground structures and resonators. In terms of design methodologies, the approaches based on artificial intelligence methods and optimizations algorithms are worth noticing.

In [25,26,27,28,29], defected ground methods were used to decrease MC between patch antennas. Although these techniques were effective in MC suppression, they also increased the complexity and the fabrication cost of antenna systems, as well as degraded the antenna radiation patterns, none of which is desirable [30,31]. Furthermore, in defected ground approaches, a sufficiently large distance between the antenna elements has to be maintained, which is detrimental from the geometrical flexibility standpoint [24].

Another type of structures that have been widely used to reduce MC between the antenna cells, especially in the MIMO structures, are resonators [32,33]. Resonators have been widely used in the structures of microwave devices to improve the functionality of the components [34,35,36]. In [37], a meandered resonator was proposed to reduce mutual coupling in the antenna array; the incorporation of the resonator resulted in shifting the antenna operating frequency. A resonator implemented using the coupled lines was used in [38] for increasing isolation between antennas. The advantage of this approach is that coupled lines can create a wide suppression band, which may increase the isolation in the structure over a broader frequency range. A resonator was incorporated in [39] with a dollar-shaped structure. The solution proposed in [39] led to a large distance between the antenna elements, which translates into an oversized overall structure. A related approach is utilization of photonic crystals, which can be applied for higher frequency operation [40,41,42,43,44,45]. This technique was recently used for designing high-isolation antenna systems [46,47].

Although the development of appropriate antenna geometry, here, oriented towards mutual coupling suppression, is essential, an appropriate adjustment of geometry parameters leading to superior performance, is just as important. Given the challenges related to simultaneous optimization of multiple parameters, often in a global sense, and at the level of costly full-wave electromagnetic (EM) analysis, efficient design methods are in high demand. Recently, artificial intelligence (AI) methods have been widely applied in solving engineering problems [48,49,50,51,52,53,54,55], especially for modeling and optimum design prediction tasks. Also, deep ANNs and deep convolutional neural networks (CNNs) have been used in engineering applications [56,57]. In particular, the AI methods have been utilized for design of microwave components [58,59,60]. In [61,62], artificial neural networks (ANNs) were used for optimization of the antenna structures. Rigorous optimization methods were utilized to design both microwave components [63,64,65,66] and antenna structures [7,67,68,69,70,71]. In [67], the swarm intelligence (SI) algorithms operations were applied in a synthesis of the linear antenna array. A hybrid particle swarm optimization (PSO) algorithm was utilized in [68] to improve the bandwidth of an inverted-F antenna. In [72], ANN modeling was used for reduction of mutual coupling in cross-dipole antenna for 2.2–2.7 GHz band. The resonator plane in [72] was applied orthogonal to the antenna planes. However, the obtained results in [72] showed that only 5.6 dB improvement was achieved in the mutual coupling reduction with the presented method. In addition, recently, automated machine learning or AutoML technique has been presented, which can provide low-code and no-code machine learning. The AutoML technique can be used as effective surrogate models to ease design and modeling of engineering problems [73,74,75].

In this paper, an artificial-intelligence-enabled technique for reducing undesirable mutual coupling in antenna arrays is proposed. Our approach employs a resonator inserted between closely spaced patch antennas, the dimensions of which are established using an inverse neural network surrogate. The inverse ANN model is identified using sampled EM-simulation data and the PSO algorithm that facilitates the surrogate training process and ensures globally optimum setup of its hyper-parameters. Furthermore, utilization of the inverse model with its inputs being the values of isolation and return loss parameters and the outputs being the geometry parameters of the resonator allow us to avoid costly EM-driven optimization of the circuit, as the optimum dimensions are directly retrieved from the surrogate. The main contributions of this paper include: (a) presenting a new resonator inserted between the antenna arrays, (b) proposing a new inverse ANN-PSO model to find the optimum values of the resonator geometry parameters and (c) presenting a new design methodology based on the design inverse model, which can also be used to ease the design processes of the other similar microwave devices. The presented approach is demonstrated to significantly improve antenna element isolation (by 30 dB) for an exemplary setup of two patch antennas operating at 2.45 GHz.

## 2. Antenna Isolation Improvement Using Resonators and Ai-Based Dimensioning

This section introduces the proposed approach to mutual-coupling reduction in antenna arrays using resonators optimized by means of artificial intelligence techniques. Section 2 outlines the overall methodology. In Section 3, we discuss a conventional antenna structure used as an illustration example. Section 4 introduces the proposed resonator circuit. The construction of the inverse ANN surrogate is elaborated on in Section 5, and Section 6 describes the isolation-enhanced array structure.

The flowchart of the proposed design method for designing of the antenna array featuring desired isolation and return loss is presented in Figure 1. As shown in the figure, the design process consists of five steps. In Step 1, a conventional antenna array without isolation enhancement is considered. In Step 2, a resonator will be introduced to be incorporated between the antenna arrays. Step 3 can be divided into three sub-steps. In Step 3.1, electromagnetic simulation data is acquired to render a raw artificial neural network (ANN) model in Step 3.2. In Step 3.3, the optimization algorithm is employed to carry out the neural network training (in particular, to assign the network weights and bias levels), thereby ensuring its sufficient predictive power and making it suitable for design purposes. Finally, in Step 4 and Step 5, the optimized inverse ANN model is utilized to design the antenna array and to achieve the required levels of return loss and mutual coupling at the target operating frequency.

## 3. Conventional Antenna Array Structure

In a conventional antenna array structure, when multiple antennas are placed close to each other, they can influence each other’s performance due to electromagnetic coupling. In a standard two-element antenna array, ideally, high isolation is desirable for ensuring that the signals received or transmitted by one antenna have minimal impact on the other antenna. High isolation helps to avoid interference between the antenna elements and enables them to operate independently. The isolation in a conventional two-element antenna array structure is not acceptable due to several factors.

The antennas in a conventional array are typically placed in close proximity to each other. When antennas are close together, the generated electromagnetic fields can interact with each other, leading to mutual coupling. The proximity increases the likelihood of coupling and reduces isolation. Also, the antennas in an array structure can scatter electromagnetic waves. When one antenna scatters waves, they can reach the neighboring antenna and couple with it, causing interference. This scattering phenomenon further reduces isolation. The mutual coupling is often frequency-dependent. The level of coupling between antennas can vary with the operating frequency, resulting in different levels of isolation at different frequencies.

Figure 2 shows the structure of the standard two-element antenna array, and Figure 3 shows the full-wave electromagnetic (EM) simulation results. It can be noted that the element isolation is only about 9 dB at the operating frequency of 2.45 GHz. The array structure is designed and implemented on Rogers RO4003C substrate. The standard antenna array works at 2.45 GHz; the corresponding guided wavelength is λ = 74.8 mm. Consequently, the absolute element distance of 5.8 mm corresponds to 0.07λ. As seen in Figure 3, the primitive antenna array without resonator provides 30 MHz bandwidth from 2.435 to 2.465 GHz, which in this band, S_11_ is less than 10 dB.

## 4. Design and Analysis of Isolation Enhancement Resonator

In order to increase the isolation between the antenna elements, an isolation-enhancement resonator is proposed, which is shown in Figure 4. The resonator includes two long high-impedance lines, and two types of long and short teeth, which are repeated along the main resonator lines. The geometry parameters of the proposed resonator, marked in Figure 4, are essential to obtaining the desired isolation and return loss for the antenna array elements.

The applied long high impedance lines in the proposed isolation-enhancement resonator act as impedance transformers. When the signal propagates through these lines, the impedance is transformed, leading to a significant change in the impedance seen by the antenna elements. By appropriately designing the impedance transformation, the resonator can effectively decouple the antenna elements, reducing the mutual coupling. Also, the utilized long and short teeth along the main resonator lines serve to reflect and absorb the electromagnetic waves. These teeth act as resonant elements, causing the incident waves to reflect and interfere with each other. This interference helps in canceling out the mutual coupling effects between the antenna elements. Moreover, the arrangement of the long and short teeth in the resonator creates a spatial distribution of currents. This distribution results in cancellation of the current components responsible for mutual coupling.

The architecture of the considered array antenna with the resonator incorporated in between the elements has been shown in Figure 5. As elaborated on in the previous section, the proposed inverse ANN surrogate can estimate the resonator parameters based on the desired values of isolation and return loss.

## 5. Inverse ANN Surrogate: Architecture and Identification

In this section, we discuss the inverse ANN surrogate employed to facilitate the optimization process of the antenna array of Section 4. As mentioned in the previous section, the appropriate adjustment of dimensions of the proposed isolation-enhancement resonator is important to ensuring the acceptable isolation level. At the same time, direct optimization of the system at the EM-simulation level of description is a computationally expensive endeavor. In order to alleviate this difficulty, an inverse ANN surrogate is proposed, which is constructed to predict the resonator parameters based on the desired values of isolation and return loss. The architecture of the proposed inverse ANN model has been illustrated in Figure 6. It can be observed that the desired values of isolation and return loss are the input arguments of the surrogate. Using these, the inverse surrogate predicts the resonator parameters, which are the length of the resonator (*L_R_*), the length of tooth 1 (*L_T_*_1_) and the length of tooth 2 (*L_T_*_2_). Also, the MATLAB R2021b software is used to simulate the proposed inverse ANN model.

In this work, among available architectures [76], we utilize a feedforward multi-layer perceptron (MLP) [77] as the underlying structure of the inverse ANN surrogate. It is the most common type of ANN, which is suitable for our purposes.

There are several reasons that MLP feedforward network is presented in the proposed approach. A feedforward MLP network is a type of ANN commonly used for modeling complex relationships and solving various machine learning tasks. It includes multiple layers of interconnected artificial neurons, where information flows in one direction, from the input layer through hidden layers to the output layer. Mutual coupling in antenna arrays often involves complex nonlinear relationships; so, MLPs are capable of modeling nonlinear relationships due to their ability to introduce nonlinearity through activation functions in the hidden layers. This enables the MLP to capture the complex dependencies between the antenna elements and the resulting coupling effects. Also, after the training process, an MLP can generalize its learned knowledge to unseen data. This means that it can accurately predict the mutual coupling effects for antenna array configurations that were not present in the training dataset. This generalization capability allows the MLP to serve as a surrogate model for mutual coupling reduction, providing accurate predictions and insights for a wide range of antenna array scenarios. Therefore, MLPs can be computationally efficient for predicting mutual coupling effects, compared to complex numerical simulations or analytical models.

As mentioned, the presented neural network contains several layers with the neurons interconnected with those of other layers. The network weights are adjusted during the training process. The most widespread training approach is the back propagation algorithm (BPA) [76], which modifies the weights of each layer of the MLP based on the approximation error of the model. However, BPAs suffer from several disadvantages, including a risk of getting stuck in a local optimum, as well as slow convergence. Here, the particle swarm optimization (PSO) algorithm [78] is chosen for optimizing the weight and the bias values of the ANN. Table 1 summarizes the parameters of the proposed inverse ANN surrogate.

PSO, developed in 1995 [78], mimics the social mechanisms, for instance, of a flock of birds (referred to as particles in the nomenclature of PSO). The PSO method is a population-based evolutionary method and has been demonstrated successful in ANN model training [79,80]. In the applied PSO algorithm for the proposed method, the two main search mechanisms include biasing of the particle relocation towards the globally best position identified by the swarm as well as locally best position remembered by each particle. PSO algorithm explores the entire search space and has the ability to escape local optima. In contrast, BPA is a local optimization algorithm that adjusts the weights based on the local gradient information. The PSO global optimization capability makes it suitable for finding better solutions in complex and non-convex optimization problems, such as training an MLP for mutual coupling reduction. PSO considers a balance between exploration and exploitation during the optimization process. It uses both the personal best and global best information to update the particles positions, allowing them to explore the search space, while also exploiting the information from the best-performing particles. This ability helps PSO in effectively searching for optimal solutions and avoiding premature convergence. In contrast, The BPA local gradient-based updates may sometimes lead to convergence to suboptimal solutions.

The flowchart of the proposed inverse ANN model is illustrated in Figure 7. At first step, the structure of the network, the hidden layer numbers and the neuron numbers are considered due to the input and output parameters and the problem conditions. In the proposed inverse ANN model, by changing the input and output parameters, a new ANN model is presented in which the output parameters of the antenna are the input of the ANN model. This method can significantly help the designers to design the antenna with desired output parameters using the proposed inverse ANN model. In the next steps, the weights and biases of the inverse ANN are defined randomly, and then, as explained in the flow chart, these values will be calculated by using the PSO optimization algorithm. Finally, after reaching desired accuracy, the inverse ANN model will be created. In addition, the pseudo-code algorithm of the presented method is described in Algorithm 1.
**Algorithm 1** Pseudo-code of the presented method1:Load data as Excel file; Data = xlsread (‘Data_Antenna.xlsx’)2:Set X and Y as inputs and outputs vectors3:Normalize the input and output vectors as XN and YN4:Set test and train Data with desired ratio as Xtr, Xts, Ytr, and Yts5:Define feed forward inverse ANN network and define activation functions and network parameters6:Call TrainUsing_PSO function to find the weight and biases of the feed forward inverse ANN network7:Set best cost function as output of function and Xtr and Ytr as inputs8:Define total number of undefined parameters of wights and biases in the inverse ANN network TotalNum = IW_Num + LW_Num + b1_Num + b2_Num9:Set swarm size and maximum iteration as 500 and 2000 for the PSO algorithm10:Define cognition coefficient as C1 = 2 and social coefficient as C2 = 4 − C111:For *p* = 1: SwarmSize12:Initialize the particle Position, Cost, Velocity, Best.Position and Best.Cost13:For *p* = 1: MaxIteration14:Update the particle Position, Cost, Velocity, Best.Position and Best.Cost15:For each particle, compare its new objective function value with its personal best value16:If the new value is lower, update the personal best position and value accordingly17:Compare the objective function value of each particle with the current global best value18:If a particle value is lower, update the global best position and value accordingly19:Return the best cost function and best values of weights and biases20:Simulate the trained network with train data and obtain the train output data of network YtrNet = sim(Network, Xtr);21:Simulate the trained network with test data and obtain the test output data of network YtsNet = sim(Network, Xts)22:Calculate the MRE error for train output data of networkMREtr = mean(abs((Ytr − YtrNet)/Ytr))23:Calculate the MRE error for test output data of networkMREts = mean(abs((Yts − YtsNet)/Yts))

Figure 8 illustrates the training process and predictive power of the inverse ANN. The training data has been acquired using ADS 2022 software and electromagnetic simulations. Subsequently, the data is used to train the presented model as an inverse ANN surrogate. It can be observed that the model is capable of predicting the resonator dimensions with the accuracy acceptable for design purposes. The summary of the modeling errors has been provided in Table 2.

## 6. Antenna Array with Mutual Coupling Reduction

In this section, we consider two specific array examples illustrating the design methodology introduced in the earlier sections.

### 6.1. Example I

In this example, the design goals concerning the required values of element isolation and matching, are set to |*S*_21_| = −30.7 dB, and |*S*_11_| = −33.1 dB. Figure 9 shows a comparison between the performance parameters predicted by the inverse ANN surrogate and the EM-simulated ones. The agreement between the two data sets is excellent, demonstrating that the inverse model is indeed a relevant design tool. The isolation and return loss, obtained by using EM simulation, based on the predicted values by inverse ANN model are |*S*_21_| = −29.2 dB and |*S*_11_| = −30.4 dB, respectively.

### 6.2. Example II

In the second example, the higher attenuations of |*S*_21_| = −44.8 dB and |*S*_11_| = −24.3 dB are used as design goals. The obtained results demonstrate that the inverse ANN surrogate can predict the resonator parameters even for more demanding performance requirements. The isolation and return loss, obtained by using EM simulation, based on the predicted values by inverse ANN model are |*S*_21_| = −46.2 dB and |*S*_11_| = −26 dB, respectively. Also, as shown in Figure 10, the performance parameters of the array using the data predicted by the inverse NN surrogate are better than the target, which corroborates suitability of the inverse ANN model for the considered task. As seen in Figure 10, the proposed antenna array with resonator provides 50 MHz bandwidth from 2.425 to 2.475 GHz, which in this band S_11_ is less than 10 dB. The result shows 66% improvement in operating bandwidth, due to the proposed resonator.

The current distribution within the antenna array structure has been depicted in Figure 11. As it can be seen, the proposed structure can effectively suppress the coupling between the antenna elements at the operating frequency. In Figure 11a, two antenna patches are applied close to each other, and as can be seen, there is mutual coupling between two arrays. In this case, mutual coupling has occurred and the electric and magnetic fields of one patch affect the other patch, resulting in a change in the current distribution. This can lead to an uneven distribution of current across the patches, which can impact the radiation pattern and efficiency of the antenna array. In Figure 11b, the proposed resonator is applied between two patches. As seen, the mutual coupling is significantly reduced and the current distribution in the antenna array is improved. The proposed resonator acts as a filter that reduces the coupling between the patches.

Gain parameters of the designed antenna array with and without the isolation-enhancement resonator are depicted in Figure 12. It can be seen that the gain parameter of the proposed antenna array with the isolation-enhancement resonator is 6.15 dB at 2.45 GHz, while this parameter is about 5.95 dB for the simple two-element antenna array without resonator at this frequency. The results show that applied isolation-enhancement resonator increases the gain parameter. Figure 13 shows the radiation patterns of the proposed antenna array with and without the isolation-enhancement resonator at 2.45 GHz, when port 1 of the antenna is excited. In the main lobe patterns of the upper-sphere space, results are basically consistent.

The antenna array considered in Example II has been fabricated and experimentally validated. Figure 14 shows the antenna prototype, as well as measured frequency response. The obtained results fully corroborate predictions of the proposed inverse ANN model. In particular, the resonator with dimension rendered by inverse ANN does reduce undesirable coupling upon being inserted between the two radiators. The experimental results show −38 dB and −19 dB attenuation levels in magnitudes of S_12_ and S_11_ parameters, respectively.

A comparison between the proposed structure and state-of-the-art designs reported in the literature have been listed in Table 3. It can be concluded from the data therein that the performance of the proposed design is competitive over the benchmark, especially in terms of the achieved mutual coupling reduction between the antenna elements.

## 7. Conclusions

In this paper, a new approach to reducing undesirable coupling in antenna arrays, using dedicated resonators and surrogate modeling, has been proposed. The presented methodology capitalizes on inverse ANN models that directly render optimum values of the resonator geometry parameters based on the input data being the desired levels of isolation and the return loss parameters. The inverse model is based on a feedforward MLP architecture, trained using nature-inspired algorithms (here, PSO), which ensures a globally optimum setup of the model hyper-parameters. For the sake of illustration, two design cases are studied, both concerning antennas that operate 2.45 GHz with 0.07λ edge-to-edge distance. In both cases, the obtained results demonstrate good agreement between surrogate model prediction and EM simulation. At the same time, the performance of the designed arrays is superior over the benchmark in terms of the achieved mutual coupling reduction. In addition, in future works, the applicability of the proposed approach to other frequencies and antenna types, such as higher frequency bands or different antenna geometries, will be studied to assess the versatility and generalizability of the technique. Also, the feasibility of extending the proposed method to address mutual coupling reduction in multi-band or wideband antenna arrays will be investigative. Moreover, in the future works, hybrid optimization techniques that combine PSO with other algorithms will be performed to improve optimization efficiency and enhance the likelihood of finding global optima.

## Figures and Tables

**Figure 1 sensors-23-07089-f001:**
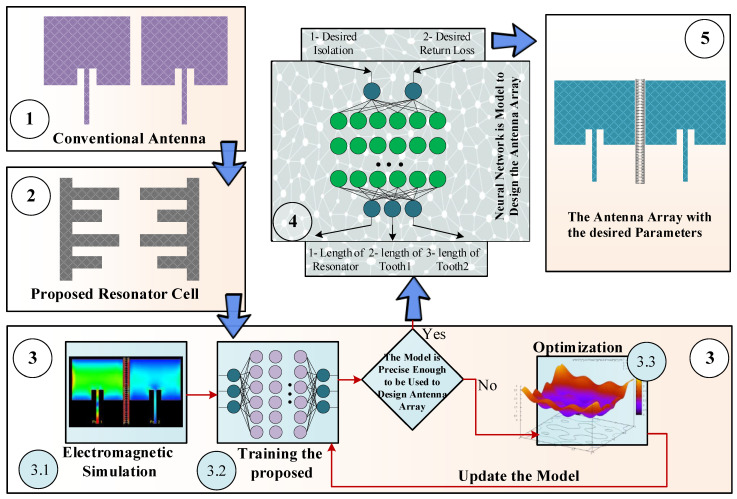
The flowchart of the proposed antenna array design methodology involving inverse ANN surrogates and numerical optimization. The framework allows for rendering antennas that operate at the desired frequency and features the required levels of mutual coupling and return loss.

**Figure 2 sensors-23-07089-f002:**
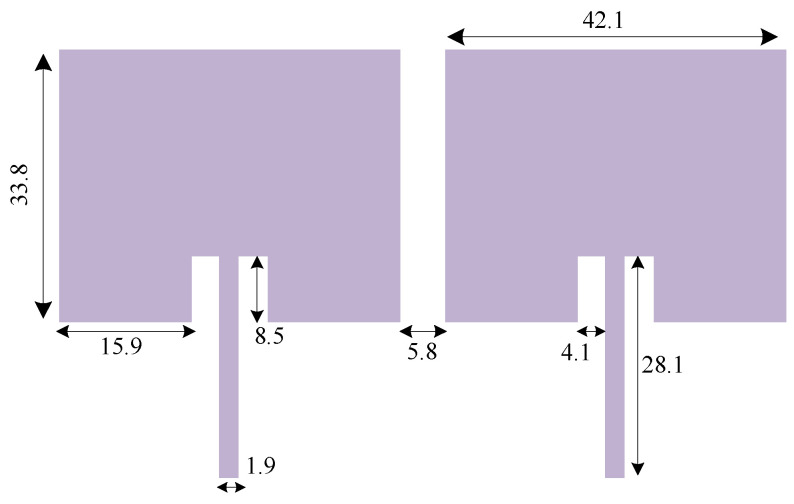
Geometry of the typical two-element antenna array with 0.07λ distance between two patch antennas (*λ* being the wavelength corresponding to the operating frequency of 2.45 GHz). All dimensions are indicated in millimeter (mm) unit in the antenna array layout.

**Figure 3 sensors-23-07089-f003:**
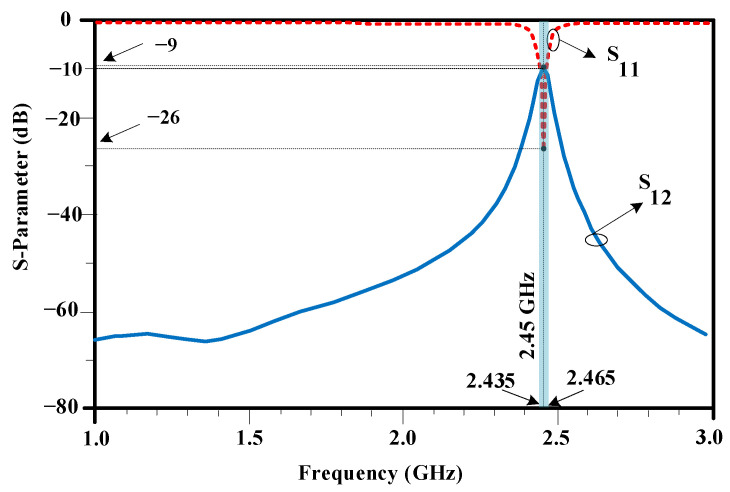
EM-simulated *S*-parameters of the standard two-element antenna array with 0.07λ distance between two patch antennas. In this figure, the solid line indicates the S_12_ parameter, which indicates the isolation factor between two antenna arrays. Also, the dashed line indicates the S_11_ parameter, which shows the return loss of the antenna array.

**Figure 4 sensors-23-07089-f004:**
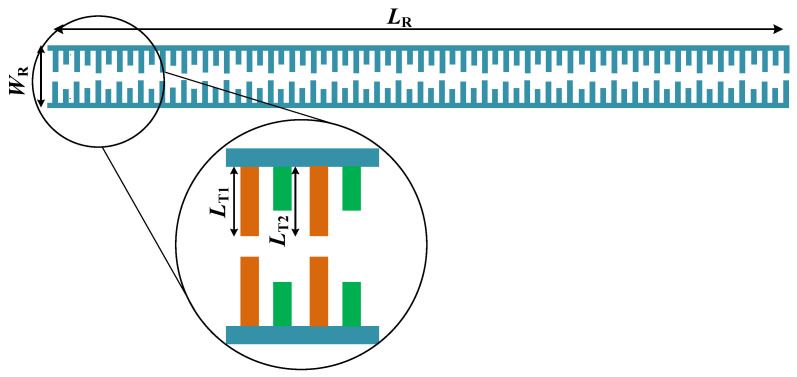
Parameterized geometry of the proposed isolation-enhancement resonator. The teeth lengths of the proposed isolation-enhancement resonator are shown with *L*_T1_, and *L*_T2_, which are corresponding to tooth 1 and tooth 2 in the resonator, respectively.

**Figure 5 sensors-23-07089-f005:**
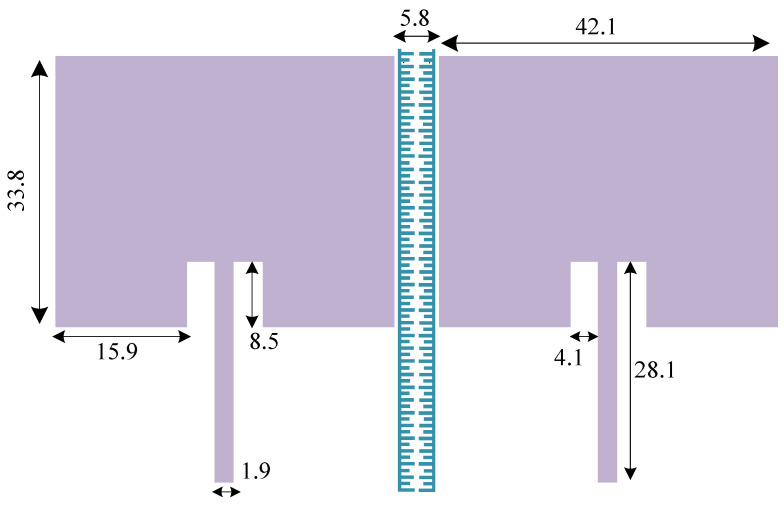
The structure of the considered antenna array with the isolation-enhancement resonator incorporated into the system. All dimensions are indicated in millimeter (mm) units in the antenna array layout.

**Figure 6 sensors-23-07089-f006:**
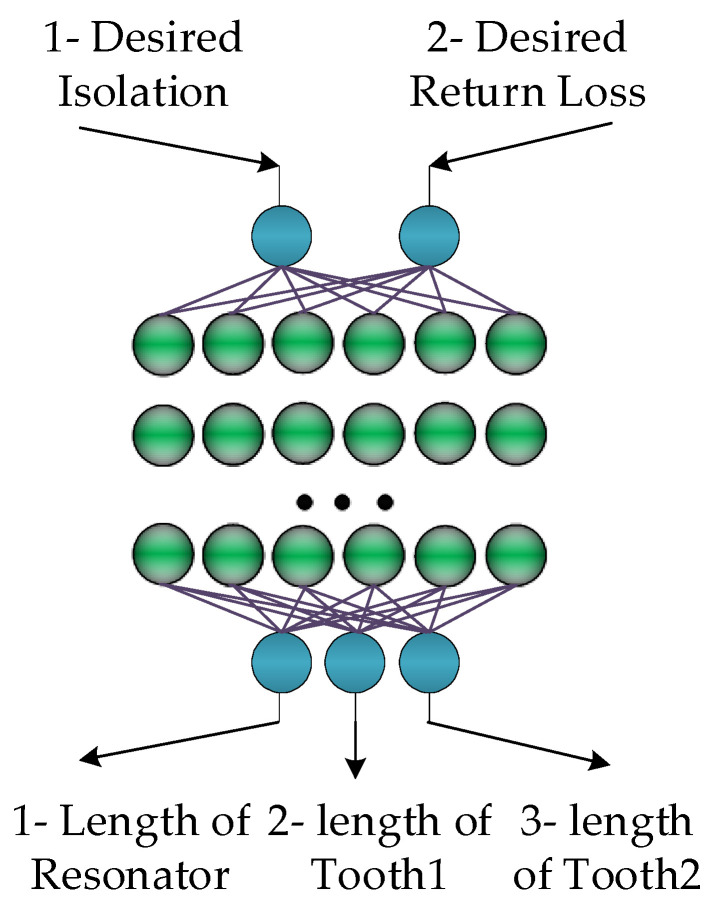
Architecture of the proposed inverse ANN surrogate. Desired isolation and desired return loss are the input parameters of the network; while, the length of resonator, length of tooth 1 and length of tooth 2 are considered as the network output parameters.

**Figure 7 sensors-23-07089-f007:**
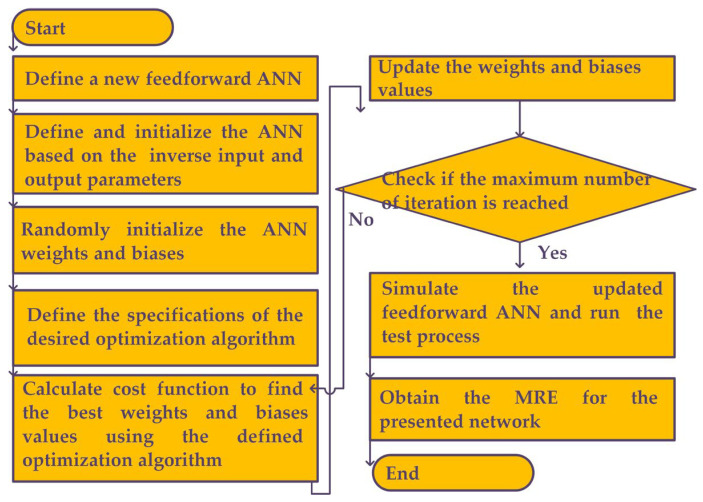
The flowchart of the proposed inverse ANN model.

**Figure 8 sensors-23-07089-f008:**
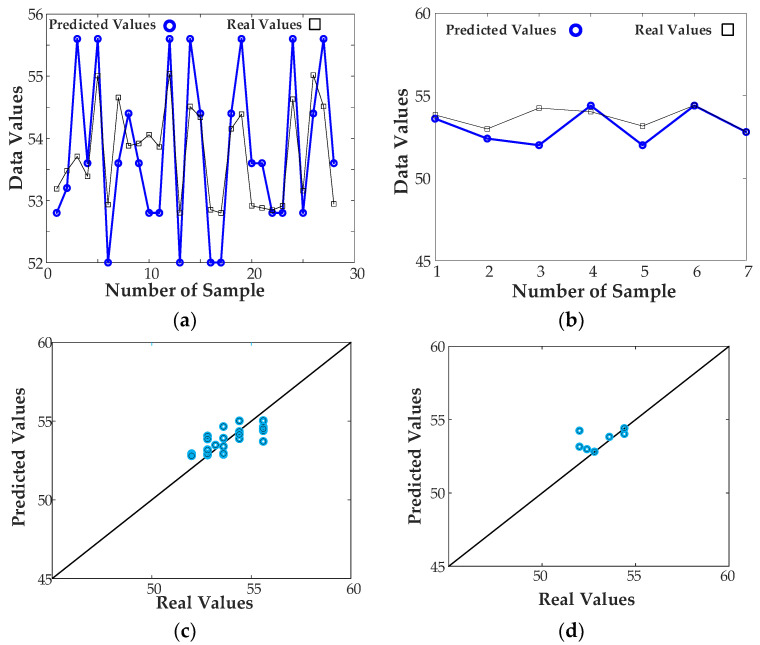
The training process and performance of the inverse ANN surrogate: (**a**) approximation performance of the model (comparison between the model-predicted and actual data at training points), (**b**) generalization performance of the model (comparison between the model-predicted and actual data for the testing data), (**c**) scatter plot (model approximation), (**d**) scatter plot (model generalization), (**e**) Evolution of the MSE error during inverse ANN model identification.

**Figure 9 sensors-23-07089-f009:**
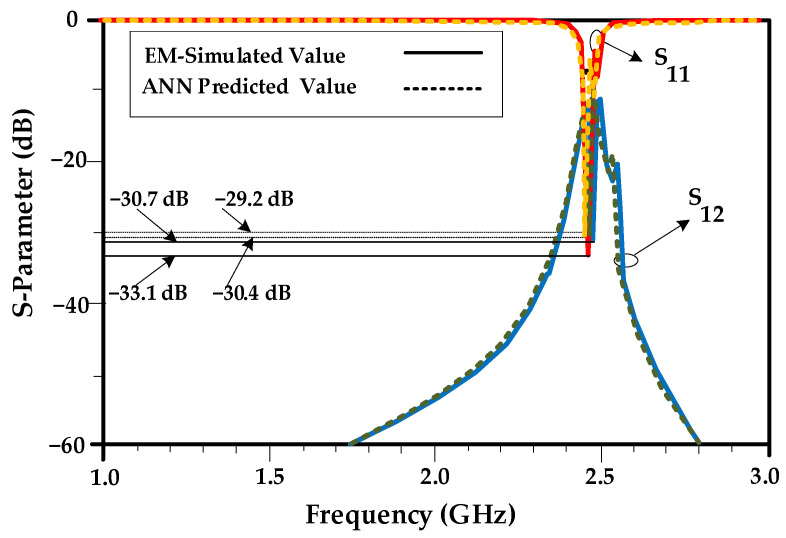
The array characteristics predicted using the inverse ANN surrogate and EM simulation (Example I). In this design example, the actual resonator parameters are *L*_R_ = 52.4 mm, *W*_R_ = 4.7 mm, *L*_T1_ = 1.8 mm and *L*_T2_ = 1.2 mm, and the values predicted by inverse ANN surrogate are *L*_R_ = 52.98 mm, *W*_R_ = 4.7 mm, *L*_T1_ = 1.806 mm and *L*_T2_ = 1.201 mm, which demonstrates the prediction accuracy of the proposed surrogate model. The shown resonator parameters are obtained through electromagnetic simulation.

**Figure 10 sensors-23-07089-f010:**
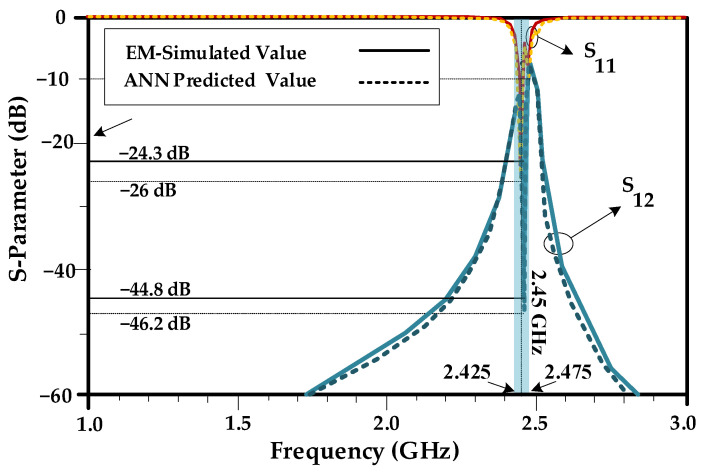
The array characteristics predicted using the inverse ANN surrogate and EM simulation (Example II). Here, the actual resonator parameters are *L*_R_ = 54.4 mm, *W*_R_ = 4.7 mm, *L*_T1_ = 1.6 mm and *L*_T2_ = 1.2 mm, and the values predicted by inverse ANN surrogate are *L*_R_ = 54.33 mm, *W*_R_ = 4.7 mm, *L*_T1_ = 1.606 mm and *L*_T2_ = 1.197 mm, which again demonstrates the prediction accuracy of the proposed surrogate model. The shown resonator parameters are obtained through electromagnetic simulation.

**Figure 11 sensors-23-07089-f011:**
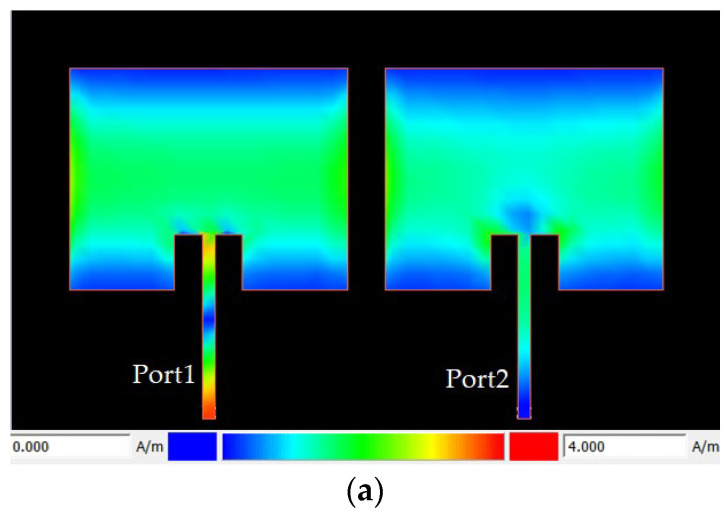
Current distribution of the designed antenna array: (**a**) a baseline two-element antenna array, (**b**) proposed antenna array with the isolation-enhancement resonator.

**Figure 12 sensors-23-07089-f012:**
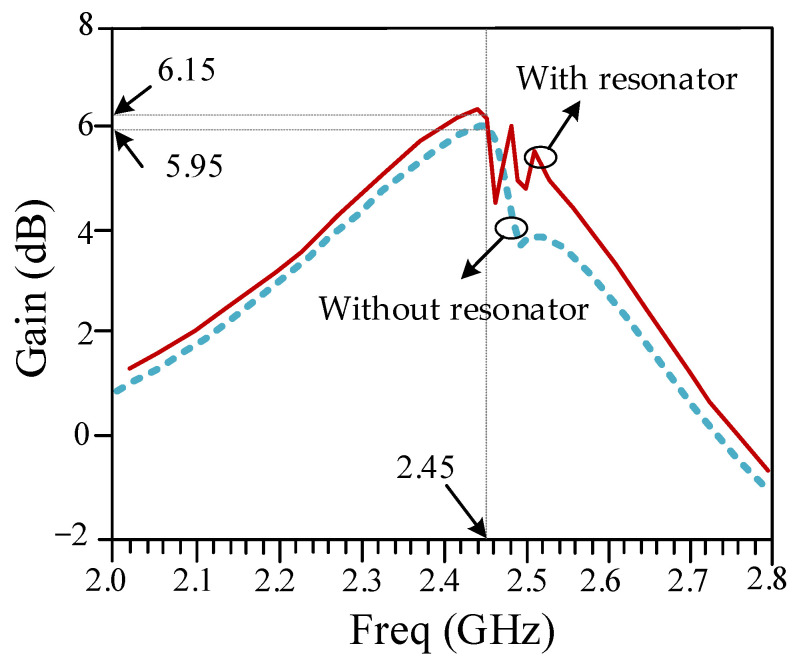
Gain parameter of the designed antenna array with and without the isolation-enhancement resonator.

**Figure 13 sensors-23-07089-f013:**
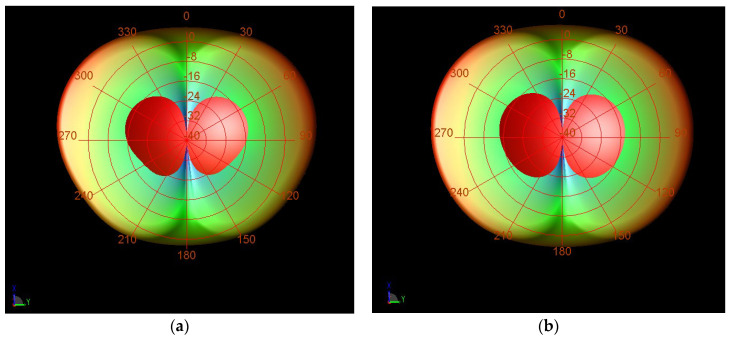
Simulated far-field radiation patterns of the proposed antenna array (**a**) without- and (**b**) with the isolation-enhancement resonator at 2.45 GHz.

**Figure 14 sensors-23-07089-f014:**
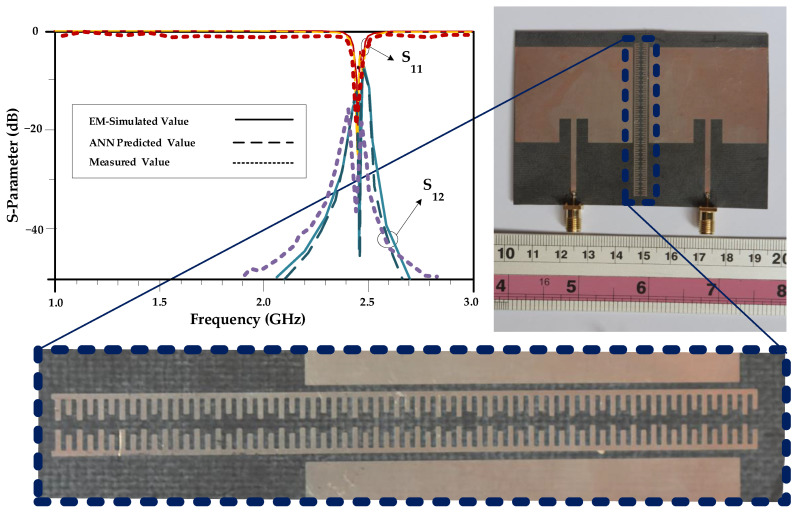
Fabricated prototype of the antenna array (Example II) and its measured frequency response. The output S-parameters of the proposed antenna array is shown on the top left of the figure, which shows the simulated and measured S_12_, isolation, and S_11_, return loss, parameters.

**Table 1 sensors-23-07089-t001:** The summarized parameters of the proposed inverse ANN surrogate.

Parameters	Details
Neural network	Feed Forward
Training algorithm	PSO
Number of neurons in the input layer	2
Number of neurons in the hidden layer	10-10
Swarm particles in PSO	500
Number of neurons in the output layer	3
Number of iterations or generations	2000
Activation function	tansig

**Table 2 sensors-23-07089-t002:** Predictive power of the inverse ANN surrogate.

Error	Network Output					
	*L*_T2_ (mm) Test		*L*_T1_ (mm)		*L*_R_ (mm)	
	Testing Data	Training Data	Testing Data	Training Data	Testing Data	Training Data
11.8512	8.5994	4.2422	3.3419	1.2651	1.2824	MRE

**Table 3 sensors-23-07089-t003:** Performance comparison between the proposed structure and state-of-the-art designs reported in the literature.

Ref. No	Freq. (GHz)	Approach	Edge to Edge Spacing	Improvement in S_21_	The Value of S_21_ with Resonator
[1]	3.94	I-section	16 mm	30 dB	NA
			(0.15 λ)		
[24]	4.8	Meandered-Line	7 mm	16 dB	22 dB
			(0.11 λ)		
[72]	2.2–2.7	ANN—Resonator Plane	NA	5.6 dB	25.3 dB
[81]	3.1–10.6	Meandered-Line	8 mm	1–24 dB	17 dB–40 dB
[82]	2.45	3D-Metamaterial	15 mm	18 dB	35 dB
			(0.13 λ)		
[83]	5.59	Planar EBG	22 mm	30 dB	NA
			(0.4 λ)		
[84]	5.8	Interdigital Lines	3.8 mm	24 dB	23 dB
			(0.07 λ)		
[85]	5.6–6.1	Metasurface	3 mm	8–27 dB	25 dB–40 dB
[86]	5	Split Ring Resonator	0.25 λ	10 dB	30 dB
[87]	2.8	Meandered Resonator	0.056 λ	8–10 dB	20 dB
Proposed	2.45	Proposed Resonator- Inverse ANN	6 mm	37.2 dB	46.2 dB
			(0.05 λ)		

## Data Availability

All the material and data associated with the study are presented within the article.

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
