# Peer review of "Mutual Coupling Reduction in Antenna Arrays Using Artificial Intelligence Approach and Inverse Neural Network Surrogates"

_sensors, 2023, doi:10.3390/s23167089_

Round 1

Reviewer 1 Report

This is a very sound work, with important practical applications. Often times, a large-size antenna elements are needed for an antenna array, and this in turn results to array elements being in close proximity to one another with deleterious effect on mutual coupling and overall array performance.

The work presented herein starts with a thorough assessment of the SOTA, and is followed by a custom methodology for solving this problem. The authors present solid simulation results but also implemented and measured one of their examples. The results are very promising and encouraging, and very useful to the readers.

Very nice use of English.

Reviewer 2 Report

This manuscript proposed a novel AI-based approach for mutual coupling reduction in antenna arrays, where artificial neural network (ANN) and particle swarm optimisation (PSO) were employed for the task of interest. In the proposed method, a stepped-impedance resonator was applied between the antennas to suppress their mutual coupling, the optimum parameters of which were obtained using the ANN model trained by PSO. The performance of proposed method has been validated using experimental data, with satisfactory results. Overall, the topic of this research is interesting, and the manuscript was well organised and written. The detailed comments are summarised as follows.

1.       The main innovation of this research should be clearly clarified in both abstract and introduction.

2.       Please broaden and update literature review on neural networks and its engineering applications. E.g., Automated damage diagnosis of concrete jack arch beam using optimized deep stacked autoencoders and multi-sensor fusion; Corrosion and coating defect assessment of coal handling and preparation plants (CHPP) using an ensemble of deep convolutional neural networks and decision-level data fusion.

3.       This study used PSO to optimise the network architecture and parameters during training. However, PSO is easy to fall into the local optimum, which can affect the prediction accuracy. How did the authors avoid this problem in this research?

4.       What are the performance evaluation metrics of proposed method? Multiple metrics are suggested to provide a comprehensive evaluation.

5.       A comparison with current other methods is suggested to demonstrate the superiority of proposed method.

6.       More future research should be included in conclusion part.

Reviewer 3 Report

In this paper, a new approach to reducing undesirable coupling in antenna arrays using dedicated resonators and surrogate modeling, has been proposed. 

Paper is good.

Presentation is good.

Results are well explained.

All figures are good.

Language is fine 

Reviewer 4 Report

The paper presents an approach to mitigate undesirable coupling in antenna arrays by employing custom-designed resonators and inverse surrogate modeling. The proposed method is demonstrated using two standard patch antenna cells with an edge-to-edge distance of 0.07λ, operating at 2.45 GHz. A stepped-impedance resonator is integrated between the antennas to reduce their mutual coupling. The study utilizes the inverse artificial neural network (ANN) model, trained with the particle swarm optimization (PSO) algorithm to determine the optimal resonator geometry parameters.

Overall, the paper introduces an innovative technique to address the problem of antenna coupling, and the research appears to be well-structured and well-executed. However, there are a few aspects that the authors may consider addressing to further strengthen the paper:

Comparison with Existing Techniques: While the results show substantial improvement compared to the conventional setup, it would be valuable to compare the proposed method with other existing techniques for reducing antenna coupling. This would help readers understand how the novel approach performs relative to well-established methods.

Literature review: All the following papers for AutoML should be cited and discussed: Human behavior in image-based Road Health Inspection Systems despite the emerging AutoML. J. Big Data 9(1): 96 (2022), AutoML technologies for the identification of sparse classification and outlier detection models. Appl. Soft Comput. 133: 109942 (2023), and FRAMED: An AutoML Approach for Structural Performance Prediction of Bicycle Frames. Comput. Aided Des. 156: 103446 (2023).

Impact of Resonator on Other Antenna Parameters: The paper focuses on the isolation improvement achieved through the resonator design. However, it would be beneficial to discuss the impact of the resonator on other antenna parameters, such as bandwidth, radiation pattern, and efficiency, to provide a more comprehensive understanding of the overall antenna performance.

Robustness Analysis: The authors should address the robustness of the proposed method against manufacturing tolerances and environmental variations. Antennas are often deployed in diverse conditions, and it is important to evaluate the technique's reliability under such circumstances.

In conclusion, the paper presents an effective approach to reduce undesirable coupling in antenna arrays using custom-designed resonators and inverse surrogate modeling. The combination of the inverse ANN model and PSO optimization provides a rapid and efficient design process. The experimental results demonstrate a remarkable improvement in isolation compared to the conventional setup, which highlights the potential of the proposed method in practical applications. Addressing the aforementioned suggestions would further enhance the paper's impact and contribute to the advancement of antenna design methodologies. Therefore, I recommend that this paper could be accepted after proper changes. If not, I am afraid to reject this paper.

The paper presents an approach to mitigate undesirable coupling in antenna arrays by employing custom-designed resonators and inverse surrogate modeling. The proposed method is demonstrated using two standard patch antenna cells with an edge-to-edge distance of 0.07λ, operating at 2.45 GHz. A stepped-impedance resonator is integrated between the antennas to reduce their mutual coupling. The study utilizes the inverse artificial neural network (ANN) model, trained with the particle swarm optimization (PSO) algorithm to determine the optimal resonator geometry parameters.

Overall, the paper introduces an innovative technique to address the problem of antenna coupling, and the research appears to be well-structured and well-executed. However, there are a few aspects that the authors may consider addressing to further strengthen the paper:

Comparison with Existing Techniques: While the results show substantial improvement compared to the conventional setup, it would be valuable to compare the proposed method with other existing techniques for reducing antenna coupling. This would help readers understand how the novel approach performs relative to well-established methods.

Literature review: All the following papers for AutoML should be cited and discussed: Human behavior in image-based Road Health Inspection Systems despite the emerging AutoML. J. Big Data 9(1): 96 (2022), AutoML technologies for the identification of sparse classification and outlier detection models. Appl. Soft Comput. 133: 109942 (2023), and FRAMED: An AutoML Approach for Structural Performance Prediction of Bicycle Frames. Comput. Aided Des. 156: 103446 (2023).

Impact of Resonator on Other Antenna Parameters: The paper focuses on the isolation improvement achieved through the resonator design. However, it would be beneficial to discuss the impact of the resonator on other antenna parameters, such as bandwidth, radiation pattern, and efficiency, to provide a more comprehensive understanding of the overall antenna performance.

Robustness Analysis: The authors should address the robustness of the proposed method against manufacturing tolerances and environmental variations. Antennas are often deployed in diverse conditions, and it is important to evaluate the technique's reliability under such circumstances.

In conclusion, the paper presents an effective approach to reduce undesirable coupling in antenna arrays using custom-designed resonators and inverse surrogate modeling. The combination of the inverse ANN model and PSO optimization provides a rapid and efficient design process. The experimental results demonstrate a remarkable improvement in isolation compared to the conventional setup, which highlights the potential of the proposed method in practical applications. Addressing the aforementioned suggestions would further enhance the paper's impact and contribute to the advancement of antenna design methodologies. Therefore, I recommend that this paper could be accepted after proper changes. If not, I am afraid to reject this paper.

Reviewer 5 Report

This paper presents Mutual Coupling Reduction in Antenna Arrays Using Artificial Intelligence Approach and Inverse Neural Network Surrogates.

Comments are as follows:

1.     The algorithm of the inverse artificial neural network (ANN) model, which is used for the optimum values of the resonator geometry parameters, is missing.

2.     How did the author get the data of the EM-simulation system at hand and train using the particle swarm optimization (PSO) algorithm? Can the authors give more details related to the particle swarm optimization (PSO) algorithm?

3.     In some of the sentences the authors have written ANN and in some of the sentences, the authors have mentioned inverse ANN. Can the authors give clarity on which algorithm they have used?

4.    How is surrogate modelling accelerating the design process?

5.     Why does the array not need to undergo direct EM-driven optimization? Can authors provide a strong reason for that?

6.     They must incorporate the algorithms or pseudo codes of the programming of PSO, ANN and MLP.

7.     How the authors have utilized all these algorithms such as PSO, ANN and MLP is not clear. The authors must provide some of the data/ programming.

8.     The authors must write the name of computer software like Python or anything in this manuscript.

Round 2

Reviewer 4 Report

The revised manuscript is much better. Thus, I recommend that this revised manuscript should be accepted.

Reviewer 5 Report

Thank you very much for your detailed responses. I am glad that my comments have been considered and that the overall quality of the paper has been improved to a level which is suitable for publication.